# DISENTANGLING THE MECHANISMS BEHIND IMPLICIT REGULARIZATION IN SGD

**Zachary Novack**
UC San Diego
znovack@ucsd.edu

**Simran Kaur**
Princeton University
skaur@princeton.edu

**Tanya Marwah**
Carnegie Mellon University
tmarwah@andrew.cmu.edu

**Saurabh Garg**
Carnegie Mellon University
sgarg2@andrew.cmu.edu

**Zachary Lipton**
Carnegie Mellon University
zlipton@andrew.cmu.edu

## ABSTRACT

A number of competing hypotheses have been proposed to explain *why* small-batch Stochastic Gradient Descent (SGD) leads to improved generalization over the full-batch regime, with recent work crediting the implicit regularization of various quantities throughout training. However, to date, empirical evidence assessing the explanatory power of these hypotheses is lacking. In this paper, we conduct an extensive empirical evaluation, focusing on the ability of various theorized mechanisms to close the small-to-large batch generalization gap. Additionally, we characterize how the quantities that SGD has been claimed to (implicitly) regularize change over the course of training. By using *micro-batches*, i.e. disjoint smaller subsets of each mini-batch, we empirically show that explicitly penalizing the gradient norm or the Fisher Information Matrix trace, averaged over micro-batches, in the large-batch regime recovers small-batch SGD generalization, whereas Jacobian-based regularizations fail to do so. This generalization performance is shown to often be correlated with how well the regularized model's gradient norms resemble those of small-batch SGD. We additionally show that this behavior breaks down as the micro-batch size approaches the batch size. Finally, we note that in this line of inquiry, positive experimental findings on CIFAR10 are often reversed on other datasets like CIFAR100, highlighting the need to test hypotheses on a wider collection of datasets.

## 1 INTRODUCTION

While small-batch SGD has frequently been observed to outperform large-batch SGD (Geiping et al., 2022; Keskar et al., 2017; Masters and Luschi, 2018; Smith et al., 2021; Wu et al., 2020; Jastrzebski et al., 2018; Wu et al., 2018; Wen et al., 2020; Mori and Ueda, 2020), the upstream cause for this generalization gap is a contested topic, approached from a variety of analytical perspectives (Goyal et al., 2017; Wu et al., 2020; Geiping et al., 2022; Lee et al., 2022). Initial work in this field has generally focused on the learning rate to batch-size ratio (Keskar et al., 2017; Masters and Luschi, 2018; Goyal et al., 2017; Mandt et al., 2017; He et al., 2019; Li et al., 2019) or on recreating stochastic noise via mini-batching (Wu et al., 2020; Jastrzebski et al., 2018; Zhu et al., 2019; Mori and Ueda, 2020; Cheng et al., 2020; Simsekli et al., 2019; Xie et al., 2021), whereas recent works have pivoted focus on understanding how mini-batch SGD may *implicitly regularize* certain quantities that improve generalization (Geiping et al., 2022; Barrett and Dherin, 2020; Smith et al., 2021; Lee et al., 2022; Jastrzebski et al., 2020).

In this paper, we provide a careful empirical analysis of how these competing regularization theories compare to each other as assessed by how well the prescribed interventions, when applied in the large batch setting, recover SGD's performance. Additionally, we study their similarities and differences by analyzing the evolution of the regularized quantities over the course of training.

Our main contributions are the following:

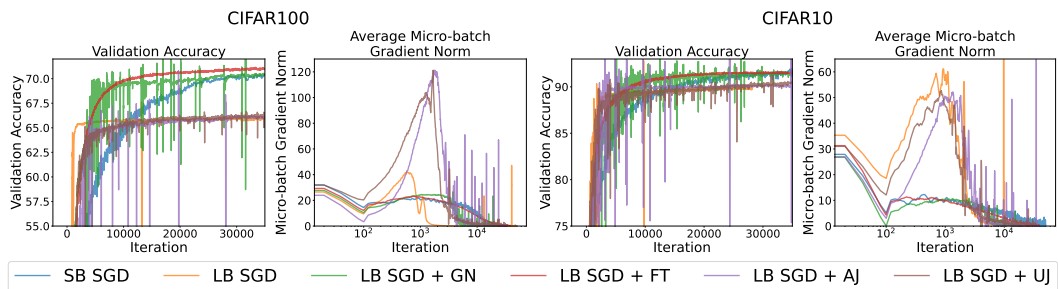

Figure 1: Validation Accuracy and Average Micro-batch ($|M| = 128$) Gradient Norm for CI-FAR10/100 Regularization Experiments, averaged across runs (plots also smoothed for clarity). In both datasets, Gradient Norm (GN) and Fisher Trace (FT) Regularization mimic the average micro-batch gradient norm behavior of SGD during early training and effectively recover generalization performance (within a small margin of error), whereas both Average and Unit Jacobian (AJ and UJ) fail to do so.

1. By utilizing micro-batches (i.e. disjoint subsets of each mini-batch), we find that explicitly regularizing either the average micro-batch gradient norm (Geiping et al., 2022; Barrett and Dherin, 2020) or Fisher Information Matrix trace (Jastrzebski et al., 2020) (equivalent to the average gradient norm when labels are drawn from the predictive distribution, detailed in Section 2.2) in the large-batch regime fully recovers small-batch SGD generalization performance, but using Jacobian-based regularization (Lee et al., 2022) fails to recover small-batch SGD performance (see Figure 1).

2. We show that the generalization performance is strongly correlated with how well the trajectory of the average micro-batch gradient norm during training *mimics* that of small-batch SGD, but that this condition is not necessary for recovering performance in some scenarios. The poor performance of Jacobian regularization, which enforces either uniform or fully random weighting on each class and example (see Section 2.3), highlights that the beneficial aspects of average micro-batch gradient norm or Fisher trace regularization may come from the loss gradient's ability to adaptively weight outputs on the per example and per class basis.

3. We demonstrate that the generalization benefits of both successful methods no longer hold when the micro-batch size is closer to the actual batch size. We too subsequently show that in this regime the average micro-batch gradient norm behavior of both previously successful methods differs significantly from the small-batch SGD case.

4. We highlight a high-level issue in modern empirical deep learning research: Experimental results that hold on CIFAR10 do not necessarily carry over to other datasets. In particular, we focus on a technique called *gradient grafting* (Agarwal et al., 2020), which has been shown to improve generalization for adaptive gradient methods. By looking at its behavior for normal SGD and GD, we show that gradient grafting recovers small-batch SGD generalization's performance on CIFAR10 but fails in CIFAR100, arguing that research in this line should prioritize experiments on a larger and diverse range of benchmark datasets.

## 2 PRIOR WORK AND PRELIMINARIES

In neural network training, the choice of batch size (and learning rate) heavily influence generalization. In particular, researchers have found that opting for small batch sizes (and large learning rates) improve a network's ability to generalize (Keskar et al., 2017; Masters and Luschi, 2018; Goyal et al., 2017; Mandt et al., 2017; He et al., 2019; Li et al., 2019) . Yet, explanations for this phenomenon have long been debated. While some researchers have attributed the success of small-batch SGD to gradient noise introduced by stochasticity and mini-batching (Wu et al., 2020; Jastrzebski et al., 2018; Zhu et al., 2019; Mori and Ueda, 2020; Cheng et al., 2020; Simsekli et al., 2019; Xie et al., 2021), others posit that small-batch SGD finds "flat minima" with low non-uniformity, which in turn boosts generalization (Keskar et al., 2017; Wu et al., 2018; Simsekli et al., 2019). Mean-

while, some works credit the implicit regularization of quantities such as loss gradient norm, the Jacobian norm (i.e., the network output-to-weights gradient norm), and the Fisher Information Matrix (Geiping et al., 2022; Barrett and Dherin, 2020; Smith et al., 2021; Lee et al., 2022; Jastrzebski et al., 2020).

Recent works have shown that one can recover SGD generalization performance by training on a modified loss function that regularizes large loss gradients (Geiping et al., 2022; Barrett and Dherin, 2020; Smith et al., 2021). While Smith et al. (2021) and Barrett and Dherin (2020) expect that training on this modified loss function with large micro-batch sizes will be unable to recover SGD generalization performance, we empirically verify this is the case.

To our knowledge, we are the first to introduce the "micro-batch" terminology to denote component disjoint sub-batches used in accumulated mini-batch SGD. This choice was made to avoid overloading the term "mini-batch" and thus clarify the work done by Smith et al. (2021) and Barrett and Dherin (2020). Note that here and for the rest of this paper, we use *Large-Batch* SGD as an approximation for full-batch GD due to the computational constraints of using the full training set on each update. We emphasize that throughout this paper, micro-batches are not meant as any sort of "alternative" to mini-batches, as they are purely an implementation feature of gradient accumulation-based large-batch SGD. We additionally leverage the work done by Agarwal et al. (2020), who propose the idea of *grafting* as a meta-optimization algorithm, though in the paper their focus mostly rests on grafting adaptive optimization algorithms together, not plain mini-batch SGD.

As a whole, our paper is situated as a comparative analysis of multiple proposed regularization mechanisms (Geiping et al., 2022; Barrett and Dherin, 2020; Smith et al., 2021; Lee et al., 2022; Jastrzebski et al., 2020) in a side-by-side empirical context, with additional ablations over how minor design choices may affect the efficacy of these proposed methods to close the generalization gap. We now discuss various implicit and explicit regularization mechanisms in more depth.

**Setup and Notation**  We primarily consider the case of a softmax classifier $f : \mathbb{R}^d \to \mathbb{R}^C$ (where $C$ is the number of classes) parameterized by some deep neural network with parameters $\boldsymbol{\theta}$. We let $\ell(\mathbf{x}, y; \boldsymbol{\theta})$ denote the standard cross entropy loss for example $\mathbf{x}$ and label $y$, and let $\mathcal{L}_{\mathcal{B}}(\boldsymbol{\theta}) = \frac{1}{|\mathcal{B}|} \sum_{(\mathbf{x},y) \in \mathcal{B}} \ell(\mathbf{x}, y; \boldsymbol{\theta})$ denote the average loss over some batch $\mathcal{B}$. Note that throughout this paper, the terms "batch" and "mini-batch" are used interchangeably to refer to $\mathcal{B}$.

## 2.1 Average Micro-batch Gradient Norm Regularization

As proposed by Smith et al. (2021), we attempt to understand the generalization behavior of mini-batch SGD by how it implicitly regularizes the norm of the **micro-batch** gradient, $\|\nabla \mathcal{L}_M(\boldsymbol{\theta})\|$ for some micro-batch $M \subseteq \mathcal{B}$. In large-batch SGD, we accomplish this through *gradient accumulation* (i.e. accumulating the gradients of many small-batches to generate the large-batch update), and thus can add an explicit regularizer (described in Geiping et al. (2022)) that penalizes the *average* micro-batch norm. Formally, for some large-batch $\mathcal{B}$ and component disjoint micro-batches $M \subseteq \mathcal{B}$, let $\nabla_{\boldsymbol{\theta}} \mathcal{L}_M(\boldsymbol{\theta}) = \frac{1}{|M|} \sum_{(\mathbf{x},y) \in M} \nabla_{\boldsymbol{\theta}} \ell(\mathbf{x}, y; \boldsymbol{\theta})$. The new loss function is:

$$\mathcal{L}_{\mathcal{B}}(\boldsymbol{\theta}) + \lambda \frac{|M|}{|\mathcal{B}|} \sum_{M \in \mathcal{B}} \|\nabla_{\boldsymbol{\theta}} \mathcal{L}_M(\boldsymbol{\theta})\|^2. \tag{1}$$

While this quantity can be approximated through finite differences, it can also be optimized directly by backpropagation using modern deep learning packages, as we do in this paper.

Note that by definition, we can decompose the regularizer term into the product of the Jacobian of the network and the gradient of the loss with respect to network output. Formally, for some network $f$ with $p$ parameters, if we let $\mathbf{z} = f(\mathbf{x}; \boldsymbol{\theta}) \in \mathbb{R}^C$ be the model output for some input $\mathbf{x}$ and denote its corresponding label as $y$, then:

$$\nabla_{\boldsymbol{\theta}} \ell(\mathbf{x}, y; \boldsymbol{\theta}) = (\nabla_{\boldsymbol{\theta}} \mathbf{z})(\nabla_{\mathbf{z}} \ell(\mathbf{x}, y; \boldsymbol{\theta})) \tag{2}$$

Where $\nabla_{\boldsymbol{\theta}} \mathbf{z} \in \mathbb{R}^{p \times C}$ is the Jacobian of the network and the second term is the *loss-output* gradient. We explicitly show this equivalence for the comparison made in section 2.3.

## 2.2 AVERAGE MICRO-BATCH FISHER TRACE REGULARIZATION

One noticeable artifact of Equation 1 is its implicit reliance on the true labels $y$ to calculate the regularizer penalty. Jastrzebski et al. (2020) shows that we can derive a similar quantity in the mini-batch SGD setting by penalizing the trace of the *Fisher Information Matrix* $\mathbf{F}$, which is given by $\text{Tr}(\mathbf{F}) = \mathbb{E}_{\mathbf{x} \sim \mathcal{X}, \hat{y} \sim p_{\boldsymbol{\theta}}(y|\mathbf{x})}[\|\nabla_{\boldsymbol{\theta}} \ell(\mathbf{x}, \hat{y}; \boldsymbol{\theta})\|^2]$, where $p_{\boldsymbol{\theta}}(y \mid \mathbf{x})$ is the predictive distribution of the model at the current iteration and $\mathcal{X}$ is the data distribution. We thus extend their work to the accumulated large-batch regime and penalize an approximation of the *average* Fisher trace over micro-batches: if we let $\widehat{\mathcal{L}}_M(\boldsymbol{\theta}) = \frac{1}{|M|} \sum_{\mathbf{x} \in M, \hat{y} \sim p_{\boldsymbol{\theta}}(y|\mathbf{x})} \ell(\mathbf{x}, \hat{y}; \boldsymbol{\theta})$, then our penalized loss is

$$\mathcal{L}_{\mathcal{B}}(\boldsymbol{\theta}) + \lambda \frac{|M|}{|\mathcal{B}|} \sum_{M \in \mathcal{B}} \|\nabla_{\boldsymbol{\theta}} \widehat{\mathcal{L}}_M(\boldsymbol{\theta})\|^2. \tag{3}$$

The only difference between the expressions in Equation 1 and Equation 3 is that the latter now uses labels sampled from the *predictive* distribution, rather than the true labels, to calculate the regularizer term. As with Equation 1, we can directly backpropagate using this term in our loss equation.

Like, in equation 2, we can decompose the regularizer term as:

$$\ell(\mathbf{x}, \hat{y}; \boldsymbol{\theta}) = (\nabla_{\boldsymbol{\theta}} \mathbf{z})(\nabla_{\mathbf{z}} \ell(\mathbf{x}, \hat{y}; \boldsymbol{\theta})) \tag{4}$$

Where the second term is another loss-output gradient.

Jastrzebski et al. (2020) observes that models with poor generalization typically show a large spike in the Fisher Trace during the early phases of training, which they coin as *Catastrophic Fisher Explosion*. In Figure 1, we show that this behavior also occurs when looking at the average Micro-Batch gradient norm.

## 2.3 JACOBIAN REGULARIZATION

Given the decompositions shown in equations 2 and 4, it is unclear in either case whether the Jacobian term is the sole source of possible generalization benefit, or if the loss-output gradient is also needed. To disentangle this effect, we borrow from Lee et al. (2022) and use the *average* and *unit* Jacobian regularization losses:

$$\mathcal{L}_{\mathcal{B}}(\boldsymbol{\theta}) + \lambda \frac{|M|}{|\mathcal{B}|} \sum_{M \subseteq \mathcal{B}} \|J_{\text{avg}}(M)\|^2, \qquad J_{\text{avg}}(M) = \frac{1}{|M|} \sum_{(\mathbf{x}, y) \in M} (\nabla_{\boldsymbol{\theta}} \mathbf{z})(\frac{1}{C} \mathbb{1}), \tag{5}$$

$$\mathcal{L}_{\mathcal{B}}(\boldsymbol{\theta}) + \lambda \frac{|M|}{|\mathcal{B}|} \sum_{M \subseteq \mathcal{B}} \|J_{\text{unit}}(M)\|^2, \qquad J_{\text{unit}}(M) = \frac{1}{|M|} \sum_{(\mathbf{x}, y) \in M} (\nabla_{\boldsymbol{\theta}} \mathbf{z})(\mathbf{u}), \tag{6}$$

where $\mathbf{u} \in \mathbb{R}^C$ is randomly sampled from the unit hypersphere (i.e. $\|\mathbf{u}\|_2 = 1$), and $\mathbf{u}$ is sampled once per micro-batch. In words, the *average* Jacobian case penalizes the Jacobian with equal weighting on every class and every example, while the *unit* Jacobian case penalizes the Jacobian with different but *random* weighting on each class and example. Note that the unit Jacobian penalty is an unbiased estimator of the Frobenius norm of the Jacobian $\|\nabla_{\boldsymbol{\theta}} \mathbf{z}\|_F^2$, which is an upper bound on its spectral norm $\|\nabla_{\boldsymbol{\theta}} \mathbf{z}\|_2^2$ (see Lee et al. (2022) for a more detailed theoretical analysis).

## 3 EXPLICIT REGULARIZATION AND GRADIENT NORM TRAJECTORY

These aforementioned explicit regularization mechanisms have previously been investigated in limited empirical settings. To the best of our knowledge, Jastrzebski et al. (2020) is the only work that has directly compared some of these regularization mechanisms, but only did so in the context of improving *small-batch* performance. Like our work, Geiping et al. (2022) is centered on the small-to-large batch generalization gap, but they do not focus *solely* on the explicit regularization they propose and do not include any analysis of the micro-batch gradient norm behavior during training. In this work, we investigate (i) how these regularizers effect generalization for an array of benchmarks and (ii) how such performance may correlate with the *evolution* of the micro-batch gradient norm during training.

Table 1: ResNet18 Test Performance for Regularizer Penalties. Values shown are average test accuracies across two to three different initializations per experiment, with corresponding confidence intervals.

| Experiment | CIFAR10 | CIFAR100 | Tiny-ImageNet | SVHN |
|---|---|---|---|---|
| SB SGD | 92.33 ($\pm$0.10) | 71.01 ($\pm$0.27) | 39.64 ($\pm$0.18) | 93.69 ($\pm$0.12) |
| LB SGD | 90.00 ($\pm$0.11) | 66.45 ($\pm$0.29) | 27.71 ($\pm$0.09) | 90.37 ($\pm$0.33) |
| GN | 91.98 ($\pm$0.03) | 70.22 ($\pm$0.27) | 37.78 ($\pm$0.07) | 92.77 ($\pm$0.01) |
| FT | 91.79 ($\pm$0.05) | 71.19 ($\pm$0.16) | 40.25 ($\pm$0.02) | 93.72 ($\pm$0.16) |
| AJ | 90.41 ($\pm$0.01) | 65.95 ($\pm$0.33) | 22.86 ($\pm$0.95) | 91.76 ($\pm$0.11) |
| UJ | 90.46 ($\pm$0.20) | 66.41 ($\pm$0.06) | 26.07 ($\pm$0.54) | 92.08 ($\pm$0.01) |

## 3.1 EXPERIMENTAL SETUP

We first focus our experiments on the case of using a ResNet-18 (He et al., 2015), with standard initialization and batch normalization, on the CIFAR10, CIFAR100, Tiny-ImageNet, and SVHN image classification benchmarks (Krizhevsky, 2009; Le and Yang, 2015; Netzer et al., 2011). Additional experiments on different architectures are detailed in Appendix A.1. Besides our small-batch ($\mathcal{B} = 128$) and large-batch ($\mathcal{B} = 5120$) SGD baselines, we also train the networks in the large-batch regime using (i) average Micro-batch Gradient Norm Regularization (GN); (ii) average Micro-batch Fisher Trace Regularization (FT); and (iii) *average* and *unit* Micro-batch Jacobian Regularizations (AJ and UJ). Note that for all the regularized experiments, we use a component micro-batch size equal to the small-batch size (i.e. 128). In order to compare the *best possible* performance within each experimental regime, we separately tune the optimal learning rate $\eta$ and optimal regularization parameter $\lambda$ independently for each regime. Additional experimental details can be found in Appendix A.5.

## 3.2 RESULTS

**(i) Average micro-batch Gradient Norm and average Fisher trace Regularization recover SGD generalization**    For CIFAR100, Tiny-ImageNet, and SVHN we find that we can fully recover small-batch SGD generalization performance by penalizing the average micro-batch Fisher trace and nearly recover performance by penalizing the average micro-batch gradient norm (with an optimally tuned regularization parameter $\lambda$, see Figure 1 and Table 1). In CIFAR10, neither penalizing the gradient norm nor the Fisher trace *completely* recovers small-batch SGD performance, but rather come within $\approx 0.3\%$ and $\approx 0.4\%$ (respectively) the small-batch SGD performance and significantly improves over large-batch SGD.

We additionally find that using the micro-batch gradient norm leads to slightly faster per-iteration convergence but less stable training (as noted by the tendency for the model to exhibit random drops in performance), while using the Fisher trace leads to slightly slower per-iteration convergence but much more stable training (see Figure 1). This behavior may be due to the Fisher trace's ability to more reliably mimic the small-batch SGD micro-batch gradient norm behavior *throughout* training, whereas penalizing the gradient norm effectively curbs the initial explosion but collapses to much smaller norm values as training progresses.

**(ii) Average and Unit Jacobian regularizations do not recover SGD generalization**    Observe in Table 1 that we are unable to match SGD generalization performance with either Jacobian regularization. In Section 2 we showed that each regularization method can be viewed as penalizing the norm of the Jacobian matrix-vector product with *some C*-dimensional vector. Crucially, both the gradient norm and Fisher trace regularizers use some form of loss-output gradient, which is data-dependent and has no constraint on the weighting of each class, while both Jacobian regularizers use data-independent and comparatively simpler vectors. Given the noticeable difference in generalization performance between the regularization methods that weight the Jacobian with the loss-output gradient and those that do not, we indicate that the loss-output gradient may be crucial to either applying the beneficial regularization effect itself and/or stabilizing the training procedure.

## 4 SHORTCOMINGS AND EXTENSIONS OF GRADIENT NORM REGULARIZATION

### 4.1 GENERALIZATION FAILURE AT LARGE MICRO-BATCH SIZES

In both successful regularization regimes, namely the average micro-batch gradient norm and average fisher trace regularizers, there is an implicit hyperparameter: the size of the micro-batch used to calculate the regularization term. Note that this hyperparameter is a practical artifact of modern GPU memory limits, as efficiently calculating higher-order derivatives for large batch sizes is not feasible in standard autodifferentiation packages. Consequently, gradient accumulation (and the use of the average micro-batch regularizer, rather than taking the norm over the entire batch) must be used on most standard GPUs (more detailed hadware specifications can be found in appendix A.5).

This restriction, however, may actually be beneficial, as Geiping et al. (2022); Barrett and Dherin (2020); Smith et al. (2021) have noted that they expect the benefits of gradient norm regularizers to break down when the micro-batch size becomes too large. To test this hypothesis, we return to the ResNet-18 in CIFAR100 and Tiny-ImageNet settings and increase the micro-batch size to as large as we could reasonably fit on a single GPU at $|M| = 2560$ in both the gradient norm and Fisher trace experiments. Additionally, we run experiments using a VGG11 (Simonyan and Zisserman, 2015) on CIFAR10, interpolating the micro-batch size from the small to large regimes. In both settings, we separately tune the learning rate $\eta$ and regularization coefficient $\lambda$ in each experiment to find the best possible generalization performance in the large micro-batch regimes.

Table 2: Test Performance for ResNet-18 with Increased Micro-Batch Size. Small-batch SGD performances: CIFAR100 = 71.01, Tiny-ImageNet = 39.64.

| Dataset | GN ($|M| = 128$) | FT ($|M| = 128$) | GN ($|M| = 2560$) | FT ($|M| = 2560$) |
|---|---|---|---|---|
| CIFAR100 | 70.22 ($\pm 0.27$) | 71.19 ($\pm 0.16$) | 64.23 ($\pm 0.49$) | 65.44 ($\pm 0.76$) |
| Tiny-ImageNet | 37.78 ($\pm 0.07$) | 40.25 ($\pm 0.02$) | 31.96 ($\pm 0.56$) | 37.71 ($\pm 0.31$) |

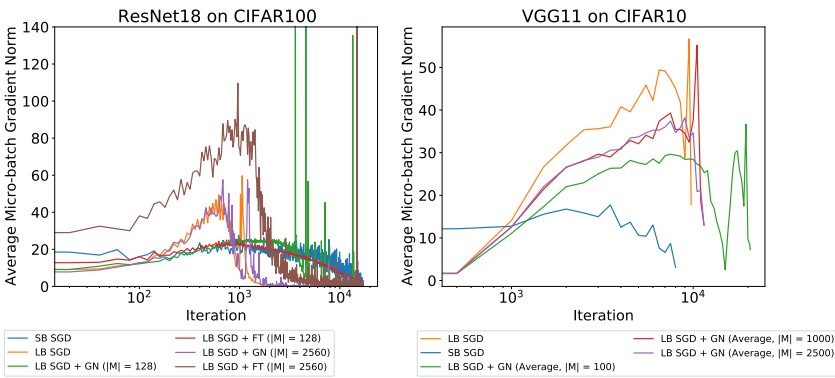

Figure 2: Average Micro-batch Gradient Norm for varying micro-batch sizes. In all experimental regimes, increasing the micro-batch size leads to a worse reconstruction of the SGD average micro-batch gradient norm behavior, especially in early training.

Table 3: Test Performance for VGG11 (no batch-normalization) in CIFAR10 with Increased Micro-Batch Size

| SB SGD | LB SGD | GN ($|M| = 100$) | GN ($|M| = 1000$) | GN ($|M| = 2500$) |
|---|---|---|---|---|
| 78.19 | 73.90 | 76.89 ($\pm 0.72$) | 75.19 ($\pm 0.10$) | 75.11 ($\pm 0.29$) |

**Results**   We successfully show that such hypotheses mentioned in Geiping et al. (2022); Smith et al. (2021); Barrett and Dherin (2020) hold true: as the micro-batch size approaches the mini-batch size,

both regularization mechanisms lose the ability to recover small-batch SGD performance (see Tables 2 and 3). Additionally, we note that using large micro-batch sizes no longer effectively mimics the average micro-batch gradient norm behavior of small-batch SGD, thus supporting our claim that matching this quantity throughout training is of key importance to recovering generalization performance (Figure 2).

## 4.2 SAMPLE MICRO-BATCH GRADIENT NORM REGULARIZATION

One potential practical drawback of these gradient-based regularization terms is the relatively high computation cost needed to calculate the second-order gradients for every component micro-batch. Instead of penalizing the average micro-batch gradient norm, we can penalize *one* micro-batch gradient norm. For some large batch $\mathcal{B}$ and fixed sample micro-batch $S$ from batch $\mathcal{B}$, we define the modified loss function

$$\mathcal{L}_{\mathcal{B}}(\boldsymbol{\theta}) + \lambda\|\nabla_{\boldsymbol{\theta}}\mathcal{L}_S(\boldsymbol{\theta})\|^2. \tag{7}$$

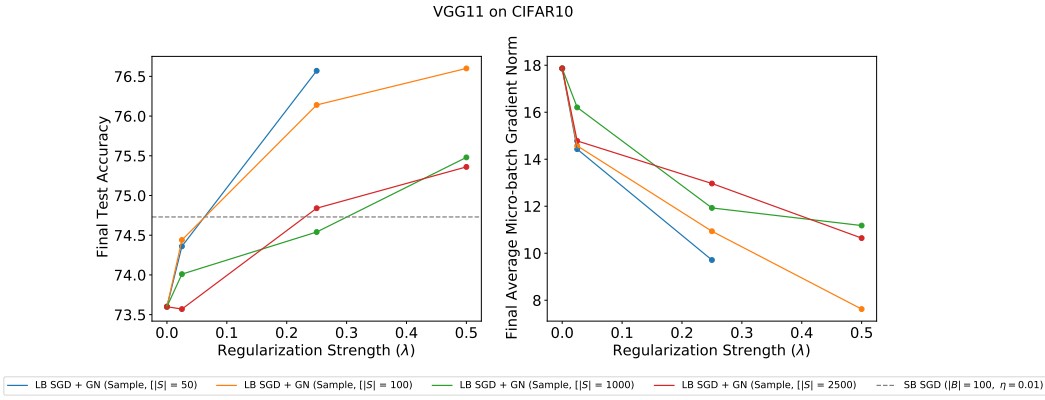

Figure 3: Explicitly regularizing the sample loss gradient norm recovers SGD test accuracy

**Results** In Figure 3, we plot the final test accuracy (left column) and the average gradient norm (right column) as a function of $\lambda$. We observe that both a larger $\lambda$ and a smaller micro-batch size $|S|$ boost test accuracy. Furthermore, we find that with the "optimal" $\lambda$ and micro-batch size $|S|$, the final test accuracy for sample micro-batch gradient norm regularization is close to (and sometimes better) than the final test accuracy for SGD. Just as we observed with the *average* Micro-batch Gradient Norm regularization, generalization benefits diminish as the sample micro-batch size approaches the mini-batch size.

## 5 IS MIMICKING SGD GRADIENT NORM BEHAVIOR NECESSARY FOR GENERALIZATION?

As seen in Figure 1, the trajectory of the average micro-batch gradient norm during training, and its similarity to that of small-batch SGD especially in the early stages of training, is strongly correlated with generalization performance. Furthermore, we have observed that models with *poor* generalization performance tend to exhibit the characteristic "explosion" during the early phase of training and quickly plummet to average micro-batch gradient norm values much smaller than seen in small-batch SGD. That being said, it is not immediately clear whether recreating the micro-batch norm trajectory of small-batch SGD is *necessary* for ensuring good generalization performance (i.e. whether good generalization directly implies gradient norm behavior similar to SGD).

To test this hypothesis, we empirically validate an orthogonal vein of optimization methods that do not explicitly regularize the micro-batch gradient norm during training for their ability to close the small-to-large batch generalization gap, and whether they too mimic the average micro-batch norm trajectory of small-batch SGD.

### 5.1 EXTERNAL AND ITERATIVE GRAFTING AND NORMALIZED GRADIENT DESCENT

Inspired by the work of Agarwal et al. (2020), we proposed to use *gradient grafting* in order to control the loss gradient norm behavior during training. Formally, for any two different optimization algorithms $\mathcal{M}, \mathcal{D}$, the grafted updated rule is arbitrarily:

$$g_{\mathcal{M}} = \mathcal{M}(\boldsymbol{\theta}_k), \qquad g_{\mathcal{D}} = \mathcal{D}(\boldsymbol{\theta}_k)$$

$$\boldsymbol{\theta}_{k+1} = \boldsymbol{\theta}_k - \|g_{\mathcal{M}}\| \frac{g_{\mathcal{D}}}{\|g_{\mathcal{D}}\|} \tag{8}$$

In this sense, $\mathcal{M}$ controls the *magnitude* of the update step and $\mathcal{D}$ controls the *direction*. We first propose **Iterative Grafting**, wherein $\mathcal{M}(\boldsymbol{\theta}_k) = \eta \nabla \mathcal{L}_M(\boldsymbol{\theta}_k)$ and $\mathcal{D}(\boldsymbol{\theta}_k) = \nabla \mathcal{L}_{\mathcal{B}}(\boldsymbol{\theta}_k)$, where $M \in \mathcal{B}$ is sampled uniformly from the component micro-batches at every update. In words, at every update step we take the large batch gradient, normalize it, and then rescale the gradient by the norm of one of the component micro-batch gradients.

Additionally, we propose **External Grafting**, where $\mathcal{M}(\boldsymbol{\theta}_k) = \eta \nabla \mathcal{L}_M(\boldsymbol{\theta}_{k'})$ and $\mathcal{D}(\boldsymbol{\theta}_k) = \nabla \mathcal{L}_{\mathcal{B}}(\boldsymbol{\theta}_k)$. Here, we use $\nabla \mathcal{L}_B(\boldsymbol{\theta}_{k'})$ to denote the gradient norm at step $k$ from a *separate small-batch SGD training run*. We propose this experiment to make a comparison with the Iterative Grafting case, since here the implicit step length schedule is independent of the current run, while with Iterative grafting the schedule depends upon the current training dynamics.

Aside from grafting algorithms, which define the implicit step length schedule at every step, we also consider the situation where the step length is fixed throughout training through **normalized gradient descent (NGD)** (Hazan et al., 2015), wherein $\mathcal{M}(\boldsymbol{\theta}_k) = \eta$ and $\mathcal{D}(\boldsymbol{\theta}_k) = \eta \nabla \mathcal{L}_{\mathcal{B}}(\boldsymbol{\theta}_k)$.

Table 4: Test Performance for Grafting / NGD Experiments

| Dataset | Model | SB SGD | LB SGD | EG | IG | NGD |
|---------|-------|--------|--------|-----|-----|-----|
| CIFAR10 | ResNet18 | 92.33 | 89.99 | 92.12 | 92.16 | 92.10 |
| | VGG16 w/Batch-Norm | 89.56 | 86.97 | 88.65 | 89.06 | 89.39 |
| CIFAR100 | ResNet18 | 71.21 | 66.17 | 68.3 | 68.4 | 66.83 |
| | VGG16 w/Batch-Norm | 64.26 | 55.94 | 59.71 | 63.48 | 58.05 |

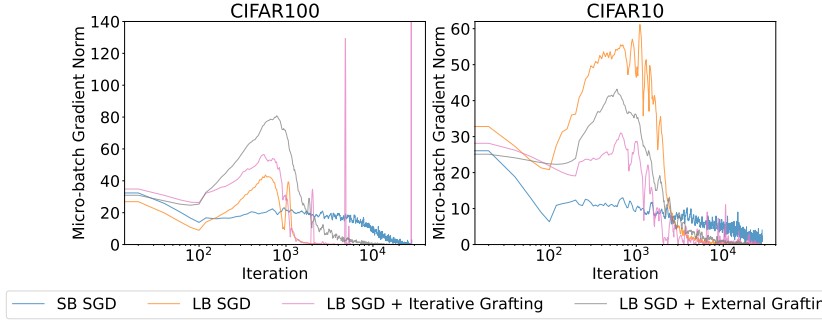

Figure 4: Average Micro-batch Gradient Norm in Grafting Experiments for ResNet-18 (Plots smoothed for clarity). In both scenarios, irrespective of generalization performance, the grafting experiments do not mimic the SGD average micro-batch gradient norm behavior.

**Results** We find that both forms of grafting and NGD can recover the generalization performance of SGD *in some model-dataset combinations* (see Table 4). Namely, though grafting / NGD seems to work quite well in CIFAR10, no amount of hyperparameter tuning was able to recover the SGD performance for either model in CIFAR100. That being said, in the CIFAR10 case we see (in Figure 4) that the grafting experiments (and NGD, not pictured) *do not* replicate the same average mini-batch gradient norm behavior of small-batch SGD despite sometimes replicating its performance. This thus gives us solid empirical evidence that while controlling the average mini-batch gradient norm behavior through explicit regularization may aide generalization, it is not the only mechanism in which large-batch SGD can recover performance.

## 5.2 WIDER IMPLICATIONS

The stark disparity in performance between the CIFAR10 and CIFAR100 benchmarks are of key importance. These differences may be explained by the much larger disparity between the mid-stage average micro-batch gradient norm behavior in the CIFAR100 case than in the CIFAR10 case (see Figure 4). This situation highlights a possible cultural issue within the deep learning community: there is a concerning trend of papers in the deep learning field that cite desired performance on *CIFAR10*, and no harder datasets, as empirical justification for any posed theoretical results (Orvieto et al., 2022; Geiping et al., 2022; Agarwal et al., 2020; Smith et al., 2021; Barrett and Dherin, 2020; Cheng et al., 2020). Given the continued advancement of state-of-the-art deep learning models, we argue that it is imperative that baselines like CIFAR100 and ImageNet are adopted as the main standard for empirical verification, so that possibly non-generalizable results (as the grafting / NGD results would have been had we stopped at CIFAR10) do not fall through the cracks in the larger community (see Appendix A.3 for more information).

## 6 DISCUSSION & CONCLUSION

In this paper, we provide a holistic account of how the proposed regularization mechanisms (Geiping et al., 2022; Barrett and Dherin, 2020; Smith et al., 2021; Lee et al., 2022; Jastrzebski et al., 2020) compare to each other in performance and gradient norm trajectory, and additionally show the limitations of this analytical paradigm for explaining the root cause of generalization. Our results with regards to the relative poor performance of the Jacobian-based regularizations somewhat conflict with the results of Lee et al. (2022), which shows positive results on using the unit Jacobian regularization with respect to improving performance *within the same batch-size regime*. We attribute this difference to the fact that Lee et al. (2022) is not concerned with cases where the small-to-large batch generalization gap exists, which is our main focus.

In light of this prior work, more research should be done to disentangle the exact effect that implicitly regularizing the loss-output gradient has on generalization performance. Next, given the success of average micro-batch gradient norm and average micro-batch Fisher trace regularization (especially with small micro-batches), future work should leverage these regularization mechanisms to investigate the possibility of ameliorating generalization, while improving time efficiency, by taking advantage of high resource, parallelizable settings. We also show that experimental findings on CIFAR10 may no longer hold in CIFAR100, which sheds light on a wider implication for the research community. Namely, we urge researchers to adapt the practice of evaluating empirical hypotheses on a more widespread, complex set of benchmarks.

We acknowledge that performance in each experiment could possibly be improved by progressively finer hyperparameter tuning, though we are confident that our core results would continue to hold in such situations given the extensive hyperparameter searches performed for each experiment. As a whole, the present research helps to shed light on the mechanisms behind SGD's generalization properties through implicit regularization, and offers robust fixes to the generalization issue at high batch-sizes.

## REPRODUCIBLITY STATEMENT

The source code for reproducing the work presented here, including all hyperparameters and random seeds, is available at `https://github.com/acmi-lab/imp-regularizers`. Additional experimental details are available in Appendix A.5.

## 7 ACKNOWLEDGEMENTS

We thank Jeremy Cohen for their useful insights and helpful discussions throughout the course of this project. TM is supported in part by CMU's Software Engineering Institute (SEI) via Department of Defense under contract FA8702-15-D-0002. SG acknowledges JP Morgan AI PhD Fellowship and Amazon Graduate fellowships for their support. ZL acknowledges Amazon AI, Salesforce Research, Facebook, UPMC, Abridge, the PwC Center, the Block Center, the Center for Machine

Learning and Health, and the CMU Software Engineering Institute (SEI) via Department of Defense contract FA8702-15-D-0002, for their generous support of ACMI Lab's research.

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

# A    APPENDIX

## A.1    ADDITIONAL REGULARIZATION EXPERIMENTS

Aside from the main results using a ResNet-18, we additionally ran the regularization experiments with a VGG11 (Simonyan and Zisserman, 2015) without batch normalization on CIFAR10. Results are shown below:

Table 5: VGG11 (no batch-normalization) Test Performance for Regularizer Penalties

| Dataset | SB SGD | LB SGD | GN | FT | AJ | UJ |
|---------|--------|--------|-------|-------|-------|-----|
| CIFAR10 | 78.19 | 73.90 | 77.62 | 79.10 | 74.09 | N/A |

Consistent with our earlier observations (see Section 3), we find that average micro-batch gradient norm and average Fisher trace regularization nearly recover SGD generalization performance, whereas average Jacobian regularization does not.

## A.2    SAMPLE MICRO-BATCH GRADIENT NORM REGULARIZATION (CONTINUED)

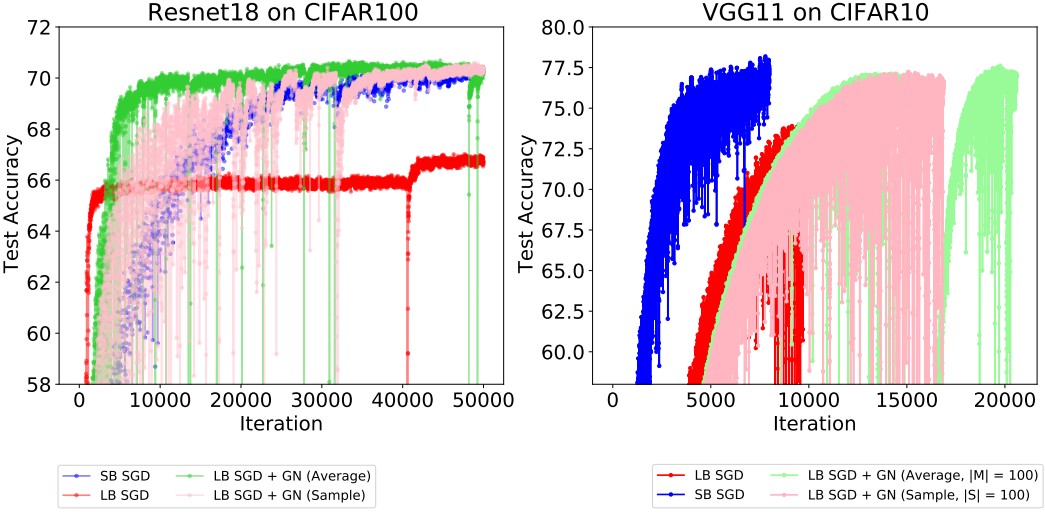

Figure 5: Explicitly regularizing the average loss gradient norm over a sample recovers SGD test accuracy

We see that switching from using the average micro-batch gradient norm to using a single sample micro-batch gradient norm as a regularizer does not impact final generalization performance. However, we do see in the CIFAR100 case that while using the sample-based regularizer is faster in terms of per iteration wall-clock time, the average-based regularizer converges to an optimal performance in considerably fewer gradient update steps (Figure 5).

## A.3    LIMITATIONS OF ANTICORRELATED PERTURBED GRADIENT DESCENT

Orvieto et al. (2022) proposes a method for improving generalization by injecting spherical Gaussian noise (with variance $\sigma^2$ as a hyperparameter) at each gradient update step that is *anticorrelated* between concurrent time steps, which they term *Anti-PGD*. They empirically show that on CIFAR10, training a ResNet18 in the large-batch regime with Anti-PGD and then shutting off the noise (to allow for convergence) allows them to beat small-batch SGD generalization performance.

However, when we extended their methodology to the CIFAR100 regime (while removing possible confounding factors such as momentum), no hyperparameter combination in the large-batch regime

was able to recover SGD generalization performance. This thus represents just one example of the possible endemic issue described in 5.2. Hyperparameter combinations and final test accuracy are show below.

Table 6: ResNet18 w/Anti-PGD on CIFAR100 (SB SGD Test Accuracy = 71.21). No hyperparameter combination comes close to recovering SGD performance.

| Learning Rate ($\eta$) | $\sigma^2$ | Test Accuracy |
|---|---|---|
| 0.5 | 0.01 | 67.54 |
| 0.5 | 0.001 | 65.44 |
| 0.1 | 0.01 | 64.55 |
| 0.1 | 0.001 | 64.90 |
| 0.05 | 0.01 | 62.52 |
| 0.05 | 0.001 | 62.78 |

## A.4 EXPLICIT REGULARIZATION WITH SOTA OPTIMIZATION TOOLS

In the present work, we are concerned with understanding the generalization performance of explicit regularization mechanisms in SGD *with no other modifications.* However, in practice many different heuristics are used to improve SGD, including momentum (Sutskever et al., 2013), weight decay (Yang et al., 2022), and learning rate scheduling (Loshchilov and Hutter, 2016). To verify that the behavior seen throughout the paper holds in more standard conditions, we return to the set-up of training a ResNet18 on CIFAR100, this time with momentum, weight decay, and cosine annealed learning rate. Here, we focus on the two successful regularization algorithms (i.e. Gradient Norm and Fisher Trace regularization). In Table 7, we see that the behavior shown in the main paper still holds: with a large gap between small-batch and large-batch SGD, Fisher Trace regularization is able to recover small-batch performance, while Gradient Norm regularization is able to beat large-batch but not fully recover small-batch performance. Specific values for the added hyperparameters are detailed in Appendix A.5.

Table 7: ResNet18/CIFAR100 Test Accuracy with momentum, weight decay, and cosine annealed learning rate. The relative performance of regularized training is maintained when adding additional optimization tools.

| SB SGD | LB SGD | LB + GN | LB + FT |
|---|---|---|---|
| 72.59 | 67.47 | 70.58 | 72.53 |

## A.5 EXPERIMENTAL SETUP DETAILS

All experiments run for the present paper were performed using the Pytorch deep learning API, and source code can be found here: `https://github.com/anon2023ICLR/imp-regularizers`.

Values for our hyperparameters in our main experiments are detailed below:

Table 8: Learning rate ($\eta$) used in main experiments

| Model/Dataset | SB SGD | LB SGD | LB + GN | LB + FT | LB + AJ | LB + UJ |
|---|---|---|---|---|---|---|
| ResNet-18/CIFAR10 | $\eta = 0.1$ | $\eta = 0.1$ | $\eta = 0.1$ | $\eta = 0.1$ | $\eta = 0.1$ | $\eta = 0.1$ |
| ResNet-18/CIFAR100 | $\eta = 0.1$ | $\eta = 0.5$ | $\eta = 0.1$ | $\eta = 0.1$ | $\eta = 0.1$ | $\eta = 0.1$ |
| ResNet-18/Tiny-ImageNet | $\eta = 0.1$ | $\eta = 0.5$ | $\eta = 0.1$ | $\eta = 0.1$ | $\eta = 0.5$ | $\eta = 0.1$ |
| ResNet-18/SVHN | $\eta = 0.1$ | $\eta = 0.1$ | $\eta = 0.1$ | $\eta = 0.1$ | $\eta = 0.1$ | $\eta = 0.1$ |
| VGG11/CIFAR10 | $\eta = 0.15$ | $\eta = 0.01$ | $\eta = 0.01$ | $\eta = 0.01$ | $\eta = 0.01$ | N/A |

Table 9: Regularization strength ($\lambda$) used in main experiments

| Model/Dataset | LB + GN | LB + FT | LB + AJ | LB + UJ |
|---|---|---|---|---|
| ResNet-18/CIFAR10 | $\lambda = 0.01$ | $\lambda = 0.01$ | $\lambda = 0.001$ | $\lambda = 0.001$ |
| ResNet-18/CIFAR100 | $\lambda = 0.01$ | $\lambda = 0.01$ | $\lambda = 5 \times 10^{-5}$ | $\lambda = 0.001$ |
| ResNet-18/Tiny-ImageNet | $\lambda = 0.01$ | $\lambda = 0.01$ | $\lambda = 1 \times 10^{-5}$ | $\lambda = 0.001$ |
| ResNet-18/SVHN | $\lambda = 0.01$ | $\lambda = 0.01$ | $\lambda = 0.0001$ | $\lambda = 0.001$ |
| VGG11/CIFAR10 | $\lambda = 0.5$ | $\lambda = 0.5$ | $\lambda = 2 \times 10^{-5}$ | N/A |

Table 10: Hyperparameters for large micro-batch experiments

| Model / Dataset | Experiment | Microbatch Size | Learning Rate ($\eta$) | Regularization Strength ($\lambda$) |
|---|---|---|---|---|
| ResNet-18/CIFAR100 | LB + GN | 2560 | 0.5 | 0.0025 |
| ResNet-18/CIFAR100 | LB + FT | 2560 | 0.1 | 0.01 |
| ResNet-18/Tiny-ImageNet | LB + GN | 2560 | 0.5 | 0.1 |
| ResNet-18/Tiny-ImageNet | LB + FT | 2560 | 0.1 | 0.1 |
| VGG11/CIFAR10 | LB + GN | 1000 | 0.01 | 0.25 |
| VGG11/CIFAR10 | LB + FT | 2500 | 0.01 | 0.25 |

**ResNet-18** For all ResNet-18 experiments, we use the standard He initialization (He et al., 2015), and the default Pytorch batch normalization initialization. Additionally, we use the standard data augmentations for CIFAR10 and CIFAR100; that is, random cropping, horizontal flipping, and whitening. For SVHN, we performed only whitening on the dataset. For Tiny-ImageNet, no data augmentations were made aside from rescaling the input images (which are $64 \times 64$) to be $32 \times 32$. Additionally, for Tiny-ImageNet the sample regularization penalties were used rather than the normal average regularization penalties given compute constraints (see Section 4.2 for the documented similarities between the sample and average regularizations).

All experiments were run for 50000 update iterations. In this case, all models are trained well past the point of reaching 100% training accuracy. No weight decay or momentum was used in *any* of the experiments. We use a large-batch size of 5120 for all experiments, and thus have 40 micro-batches of 128 examples each for the regularization experiments. We calculate the penalty term at every update step, which is different from the procedure in Jastrzebski et al. (2020), which recalculates the penalty term only every 10 update steps. For the external grafting experiments, we use the gradient norm data from a separate run of small-batch SGD with batch size equal to the same micro-batch size used for iterative grafting (i.e. 128). All experiments were run on a single RTX A6000 NVidia GPU.

For the experiments with added optimization tools in Appendix A.4, we take inspiration from Jastrzebski et al. (2020) and Geiping et al. (2022) and use momentum $= 0.9$, weight decay $= 1 \times 10^{-4}$, and a cosine annealing schedule that anneals the initial learning rate to 0 every 300 epochs. This set-up is used for all experiments in this section.

**VGG16** The set-up for the VGG16 experiments are identical to the ResNet-18 experiments, including the usage of batch normalization within the architecture.

**VGG11 without batch-normalization** For all VGG-11 experiments, we train the network with a fixed learning rate (and no momentum) until we reach 99% train accuracy. Note that we do not use any form of data augmentations. We use a small batch size of 100 and a large batch size of 5000.

Table 11: Hyperparameters for sample micro-batch experiments

| Model / Dataset | Experiment | Microbatch Size | Learning Rate ($\eta$) | Regularization Strength ($\lambda$) |
|---|---|---|---|---|
| VGG11/CIFAR10 | SB SGD | N/A | 0.01 | N/A |
| VGG11/CIFAR10 | LB + FT | 50 | 0.01 | 0.25 |
| VGG11/CIFAR10 | LB + FT | 100 | 0.01 | 0.5 |
| VGG11/CIFAR10 | LB + FT | 1000 | 0.01 | 0.5 |
| VGG11/CIFAR10 | LB + FT | 2500 | 0.01 | 0.5 |

Table 12: Hyperparameters for Grafting Experiments

| Model/Dataset | SB SGD | LB SGD | Iterative Grafting | External Grafting | NGD |
|---|---|---|---|---|---|
| ResNet-18/CIFAR10 | $\eta = 0.1$ | $\eta = 0.1$ | $\eta = 0.1$ | $\eta = 0.1$ | $\eta = 0.2626$ |
| ResNet-18/CIFAR100 | $\eta = 0.1$ | $\eta = 0.5$ | $\eta = 0.1$ | $\eta = 0.1$ | $\eta = 0.3951$ |
| VGG-16/CIFAR10 | $\eta = 0.05$ | $\eta = 0.1$ | $\eta = 0.05$ | $\eta = 0.05$ | $\eta = 0.2388$ |
| VGG-16/CIFAR100 | $\eta = 0.1$ | $\eta = 0.1$ | $\eta = 0.1$ | $\eta = 0.1$ | $\eta = 0.4322$ |

