# OpenReview forum: "Disentangling the Mechanisms Behind Implicit Regularization in SGD"
_ICLR.cc/2023/Conference — ICLR 2023 poster_

### Official Review · Reviewer_rQEx · 2022-10-21

**Confidence:** 3
**Correctness:** 3
**Technical Novelty And Significance:** 2
**Empirical Novelty And Significance:** 2
**Recommendation:** 5

**Clarity, Quality, Novelty And Reproducibility:**

I wish how this work positions in the literature could be clarified more. It could just be a clarity issue, but without better understanding this, I’m incapable of assessing the originality.

I also hope the authors could clarify about whether/how micro-batch is better than small mini-batch in practice? Is it computationally more efficient? It could just be me but I feel this part is not so clear to me.

**Strength And Weaknesses:**

Strength: using micro-batch as an estimator of the gradient norm and a regularizer is a very interesting idea. It seems to have a lot of potential.

Weakness:

(1) very little theoretical justification is provided, but the empirical results are not comprehensive enough to support the claims.

(2) if I didn’t misunderstand, one central goal of this paper is to suggest that small-batch SGD has an implicit bias that regularizes gradient norm. I’d like to better understand why this wasn’t investigated directly, but rather via an indirect route based on the extra complication of micro-batch. I’m relatively convinced that micro-batch has this implicit bias, but curious about why this serves a demonstration of the small-batch SGD story.

(3) a traditional belief is that smaller batch size leads to behaviors similar to those under larger learning rate. The implicit bias of large learning rate, in terms of regularizing the gradient, for example, has been studied already, not only empirically but also theoretically. In addition, the implicit bias of small batch size has also been studied, see e.g., [Smith et al. ICLR 2021], which was already cited. Therefore, it is not completely clear to me what’s new in terms of SGD’s implicit bias. The micro-batch part is still interesting though.


**Summary Of The Paper:**

This paper considers micro-batches, which are disjoint smaller subsets of mini-batch. Empirical evidence is provided to suggest that, although a large mini-batch does not lead to generalization as good as a small mini-batch, the introduction of micro-batches together with a large mini-batch helps recover the generalization corresponding to a small mini-batch.

**Summary Of The Review:**

Overall, I feel this paper definitely proposed an interesting idea. However, I hope to understand better what is the gain, as well as to see more evidence of its effectiveness (if not theoretically, at least empirically, and more comprehensively).

---

> ### Author Response · Authors · 2022-11-16
> **Reviewer Response (Part 1)**
>
> Thank you for your feedback! We have addressed some of your concerns below:
>
> > **I wish how this work positions in the literature could be clarified more. It could just be a clarity issue, but without better understanding this, I’m incapable of assessing the originality.**
>
> We apologize for the lack of clarity in this regard, and will attempt to clear things up here. Our work is situated within the literature as a comparative review of multiple proposed regularization mechanisms for closing the large-batch/small-batch generalization gap, which to our knowledge has not been done before. Within this, we implement the proposed regularization mechanisms side-by-side to understand how these methods compete against each other in a standardized format. Additionally, we provide ablations over some of the design choices of these regularization methods (e.g. size of micro-batch, number of micro-batches to use) that have not been previously investigated, as well as an empirical investigation of micro-batch gradient norm regularization’s explanatory power of good generalization (Section 5). We have made edits in our paper in order to make this fact more salient.
>
> > **I also hope the authors could clarify about whether/how micro-batch is better than small mini-batch in practice? Is it computationally more efficient? It could just be me but I feel this part is not so clear to me.**
>
> We apologize for the confusion, and want to clarify the relationship between micro-batches and mini-batches. For our purposes, **micro-batches** are subsets of a given batch (commonly termed as mini-batches) that only make sense to be talked about in the context of large batch training when gradients are accumulated to preserve memory. The reviewer is right in that micro-batches *can* be more computationally efficient, but this is mostly connected to the extremely data-parallel regime, where using a large batch size (made up of component micro-batches) can be more efficient and is commonly done in practice. We have modified our paper in order to clarify any confusion.
>
> > **(1) very little theoretical justification is provided, but the empirical results are not comprehensive enough to support the claims.**
>
> The authors wish to note that the high-level goal of our work is to understand how different proposed mechanisms for regularizing large-batch SGD *empirically* compare in terms of generalization performance and norm behavior. Our intention was never to provide a theoretical underpinning for why such methods could work (as this has been done extensively in previous work, see Smith et al. (2021), Geiping et al. (2022), Jastrzebski et al. (2020), Lee et al. (2022)), but rather formalize the comparison of these methods across an array of benchmarks, and also empirically understand these regularizers’ failure modes (such as increasing the micro-batch size, Section 4.1) and limitations of explanatory power (Section 5).
>
> > **(2) if I didn’t misunderstand, one central goal of this paper is to suggest that small-batch SGD has an implicit bias that regularizes gradient norm... curious about why this serves a demonstration of the small-batch SGD story.**
>
> The authors wish to clarify a few miscommunications on our part. Namely, any result with regards to micro-batches is related to **large-batch** SGD, *not* small-batch SGD (as noted above they are not really directly comparable). Additionally, the central goal of our paper is not to investigate small-batch SGD’s implicit biases in isolation (this has been done extensively previously, most similarly to our work in Geiping et al. (2022) and Jastrzebski et al. (2020)), but rather investigate *how* previously proposed hypotheses for implicit regularization actually hold up as explicit regularizers in bridging the small-batch/large-batch generalization gap. The inclusion of micro-batches (this term is coined by us, but the quantity itself appears in Barrett & Dherin (2020) and Geiping et al. (2022)) in our paper is for two reasons:
> - As noted in Geiping et al. (2022), their is theoretical justification that regularizing the *per-example* gradient norm may be beneficial for generalization; as this is computationally impractical, using micro-batches is a decent approximation of penalizing the per example for large-batch SGD.
> - Micro-batches *naturally* arise as an implementation factor of distributed training, as gradients are calculated in parallel across multiple nodes (each comprising a micro-batch), thus dramatically increasing the efficiency of large-batch training.
>
> We have made changes throughout the paper, and hope that our changes make our main contributions more clear.

---

> > ### Author Response · Authors · 2022-11-16
> > **Reviewer Response (Part 2)**
> >
> > (...continued from Part 1)
> >
> > > **(3) a traditional belief is that smaller batch size leads to behaviors similar to those under larger learning rate. The implicit bias of large learning rate, in terms of regularizing the gradient, for example, has been studied already, not only empirically but also theoretically. In addition, the implicit bias of small batch size has also been studied, see e.g., [Smith et al. ICLR 2021], which was already cited. Therefore, it is not completely clear to me what’s new in terms of SGD’s implicit bias. The micro-batch part is still interesting though.**
> >
> > We acknowledge that there are multiple competing hypotheses for why small-batch SGD generalizes better than large-batch SGD (some of which attribute generalization benefits to the implicit bias of large learning rates and/or small batch sizes). However, we emphasize that our work focuses on comparing hypotheses that attribute these generalization benefits to the implicit regularization of certain norm-based quantities.
> >
> > We emphasize that one key difference between our work and [Smith et al. ICLR 2021] is that we investigate the effect of microbatch size, and find that the benefits of explicit regularization will vanish as micro-batch size approach actual batch size. This is an important observation, because it shows that it is not enough to calculate these regularizer terms on the entire batch.
> >
> > We consider works that investigate the learning rate / batch-size relationship as a complementary but orthogonal vein of research to our work and the body of literature we draw from (Geiping et al. (2022), Jastrzebski et al. (2020), Lee et al. (2022)).

---

> > > ### Comment · Reviewer_rQEx · 2022-11-19
> > > **Thanks for your response**
> > >
> > > I've increased my rating from 3 to 5.

---

### Official Review · Reviewer_i18E · 2022-10-24

**Confidence:** 3
**Correctness:** 4
**Technical Novelty And Significance:** 3
**Empirical Novelty And Significance:** Not applicable
**Recommendation:** 6

**Clarity, Quality, Novelty And Reproducibility:**

The paper has good writing and is easy to read. Some of its results are novel, which I believe will broadly impact the community.

**Strength And Weaknesses:**

## Strength
- Though this paper is purely empirical, it provides many important intuitions for future work. For example, the failure of the Jacobian-based regularizations shows that this might not the reason for the good generalization of small-batch SGD. On the other hand, the counterexample in Figure 4 shows that we should seek other explanations beyond gradient norm regularization. The last observation that some tricks might fail on large datasets calls for verifying all the previous tricks only verified on small datasets and rethinking the importance of dataset structure.
- The paper is well structured. I enjoy the process of reading this paper.


## Weaknesses
- The performance reported in the paper is not the SOTA. I suspect the reason is that they do not use weight decay and momentum, as mentioned in the appendix. I can understand that weight decay or momentum can be regarded as some kind of regularization. Moreover, I did not find the stepsize regime. Do you use constant stepsize? In the reality, we always add weight and momentum, together with stepsize decay (or cosine learning rate) to improve the generalization. I will be very happy to increase my score if you can provide simulations showing that gradient norm regularization can also recover the generalization after adding all these common tricks.

**Summary Of The Paper:**

This paper does a thorough simulations to clarify and verify the existing hypothesis about why small batch SGD outperform its large batch counterpart. They derive many interesting results. First, they show that adding regularizer based on gradient norm or the Fisher Information Matrix trace can recover small-batch SGD generalization, while Jacobian-based regularizations fail. On the other hand, they show that the gradient norm, in some case, is not a good indicator of generalization, i.e., there exists some algorithms achieving good generalization but with large gradient norm. Last, they show that some tricks work on small dataset like CIFAR10 might fail on large dataset like CIFAR100, Anti-PGD, for example.

**Summary Of The Review:**

Please see the strength and weaknesses part.

---

> ### Author Response · Authors · 2022-11-18
> **Reviewer Response**
>
> Thank you for your review! We are glad that you enjoyed reading our paper! Below we address some of the concerns brought up.
>
> >**“The performance reported in the paper is not the SOTA… In the reality, we always add weight and momentum, together with stepsize decay (or cosine learning rate)...**
>
> We note that the main focus of our work is to disentangle how the implicit and explicit regularization of certain norm-based quantities may impact generalization in SGD. A key factor of our experimental design was thus to isolate this effect in as controlled a setting as possible (which is why we use a constant learning rate for all our experiments as the reviewer mentioned).
> The reviewer is correct in that our set up is well below SOTA performance, and this is exactly the intention, as adding external modifications like weight-decay, momentum, and learning-rate scheduling may have not allowed us to disentangle the generalization benefit of explicit regularization alone in our main results.
>
> That being said, the authors agree with the reviewer that understanding whether our initial results hold up in more SOTA settings, with all the additional modifications, is an important question. We have thus added a new section in Appendix A.4, where we train ResNet18s on CIFAR100, focusing on our successful regularization mechanisms (i.e. Gradient Norm and Fisher Trace) **with momentum, weight decay, and cosine learning rate scheduling.** In all, we observe the same trend of behavior in the simple setting – Fisher Trace regularization is able to completely recover small batch SGD generalization, and Gradient Norm is able to get well above large batch SGD performance but does not completely close the gap to recover small batch SGD performance.
>
> Additionally, we have modified the final section of the appendix (A.5) to include the optimal learning rates and regularization strength hyperparameters directly in the paper (as they only existed on the repository before) to improve overall self-cohesion.

---

### Official Review · Reviewer_fWma · 2022-10-24

**Confidence:** 4
**Correctness:** 4
**Technical Novelty And Significance:** 2
**Empirical Novelty And Significance:** 3
**Recommendation:** 6

**Clarity, Quality, Novelty And Reproducibility:**

Clarity is good.

Quality is solid.

Novelty is fine. Sections 3 and 4 might not be very novel. But Section 5 proposes a new perspective to me.

Reproducibility is good.

**Strength And Weaknesses:**

# Strength
+ The main contributions are summarized in the Summary section. Some other strength of this work includes:
+ The paper is overall very well organized. I have a good time readying most of the paper (except for a few places please see below).
+ As a scientific paper, more experiments are always welcome. With that being said, I think the conclusions presented in this work are well backed up by its experimental design. At this point I did not find a major shortcoming in terms of experimental design.
+ I really like the design of Section 5 which disentangles the effect of the magnitude and direction of the update. Such a perspective is novel to my knowledge for understanding SGD (despite it has been studied from a more theory work), and I think it may inspire a deeper understanding of SGD.


# Weakness
- A large portion of the work focuses on verifying which type of explicit regularization for large batch SGD/GD can recover the performance of small batch SGD. I would like to see some discussions on why finding the right explicit regularization (that corresponds to the implicit regularization of SGD) is important or useful.
- Note that the explicit regularization term introduces several new hyperparameters. For example the penalty strength $\\lambda$. I would like to see some discussion on how hard it is to tune the penalty strength $\\lambda$. Partly due to the new hyperparameters, I somehow feel the finding of this work might not be very useful for practitioners.
- I find Section 3.2, paragraph (ii) is very hard to interpret. Would you mind elaborate a bit? Perhaps written down some math could help.

**Summary Of The Paper:**

This work studies the implicit bias of SGD by investigating which kind of explicit regularization can help large batch SGD/GD match the performance of (small batch) SGD. Several new insights are drawn from a systematic set of experiments.

Firstly, the effect of several explicit regularizer are studied, including (1) micro-batch gradient norm regularization, (2) micro-batch fisher trace regularization, (3) micro-batch average Jacobian regularization and (4) micro-batch random and unit Jacobian regularization. It is shown that the first two regularizers are more effective than the last two.

Secondly, the work turns to looks at some extreme use cases of micro-batch gradient norm regularization: (1) when the micro batch size approaches the scale of the batch size of "large batch SGD", the regularizer no longer works well; (2) when the micro-batch gradient norm regularization is approximated by a sample micro-batch gradient norm regularization, the latter regularizer works well when he regularization strength is property turned.

Finally, the paper considered a very interesting set of experiments that involve gradient grafting, which mixes the direction and magnitude from two separate (or related) optimizers. It is shown that, for some CIFAR-10 settings, gradient grafting can recover the performance of SGD, indicating that perhaps the direction of large batch SGD update is good enough but its magnitude could have been improved. Yet, this conclusion does not carry over to harder settings, where improving the magnitude of large batch SGD update might not be sufficient for it to generalize well.

Based on the last result, an inconsistent performance of one algorithm in easy/hard deep learning application, the paper discusses some limitations of current scientific study of deep learning, where most experimental evidence is from relatively easy tasks and might not migrate to harder tasks.

**Summary Of The Review:**

Please see above. At this point I would like to vote for a weak acceptance. I will consider increasing the score based on the authors reply to my questions. I might also consider decreasing the score if I or other reviewers find a major vulnerability in the experimental design that cannot be satisfactorily addressed.

---

> ### Author Response · Authors · 2022-11-16
> **Reviewer Response (Part 1)**
>
> Thank you for the detailed response. We have attempted to respond to some of your comments below:
> >**A large portion of the work focuses on verifying which type of explicit regularization for large batch SGD/GD can recover the performance of small batch SGD. I would like to see some discussions on why finding the right explicit regularization (that corresponds to the implicit regularization of SGD) is important or useful**
>
> The authors wish to highlight that the main goal of our paper is to compare an array of proposed mechanisms for bridging the small-batch/large-batch generalization gap that regularize certain norm based quantities on a breadth of benchmarks. We note that the importance of finding a functional explicit regularization method is twofold:
> - In comparing very similar regularization mechanisms, understanding which ones perform better and where can help shed light on the training dynamics of small batch SGD. Notably, our discovery on the failure of the two Jacobian regularization methods highlights that small batch SGD may not be solely implicitly regularizing the Jacobian to achieve good generalization performance.
> - Much like how better optimization algorithms can be useful for those training large deep networks, finding the “right” explicit regularization could be of particular use to practitioners training large models in extremely data-parallel systems, where batch sizes can be quite large (like in Goyal et al., 2017) and calculation of the average micro-batch gradient norm can be done in parallel.
>
> >**Note that the explicit regularization term introduces several new hyperparameters. For example the penalty strength $\lambda$. I would like to see some discussion on how hard it is to tune the penalty strength $\lambda$. Partly due to the new hyperparameters, I somehow feel the finding of this work might not be very useful for practitioners.**
>
> The authors understand the reviewers reservations given the introduction of new hyperparameter—$\lambda$—to tune for training networks from scratch. However, we will clarify here that tuning the penalty strength $\lambda$ (in the case of the successful regularization schemes of gradient norm and fisher trace methods) was surprisingly uncomplicated. For each dataset and method, we tuned the penalty strength on a pseudo-log scale (i.e. 0.5, 0.1, 0.05, 0.01, etc.), and have mostly found the following behavior: **the optimal regularization strength (holding all other hyperparameters fixed) is generally the largest $\lambda$ value that does not lead to divergence.**  Since model divergence can be quickly assessed early in the training process, tuning $\lambda$ is a comparatively cheap hyperparameter to tune for practitioners, and only requires 1-3 full training runs to find a suitable performing $\lambda$.
>
> Furthermore, the additional hyper-parameter $\lambda$ is the weight assigned to the external regularization in the *large-batch* regime. The size of the large-batch could potentially be taken to be equal to the size of the entire training data (which in extremely parallel situations could be quite viable), in which case “batch-size” is no longer a tunable hyper-parameter.

---

> > ### Author Response · Authors · 2022-11-16
> > **Reviewer Response (Part 2)**
> >
> > (...continued from Part 1)
> >
> > >**I find Section 3.2, paragraph (ii) is very hard to interpret. Would you mind elaborate a bit? Perhaps written down some math could help.**
> >
> > We apologize for the lack of clarity in this section, and will elaborate more here:
> >
> > As we see in Table 1, both the Unit and Average Jacobian regularization methods do not recover small-batch SGD generalization performance (across all datasets). Recall that in the formulation of the Jacobian regularization (from Section 2.3), each regularization is defined as the following:
> >
> > Average Jacobian: $$\mathcal{L} _ {\mathcal{B}}(\boldsymbol\theta) + \lambda \frac{|M|}{|\mathcal{B}|}\sum_{M \subseteq \mathcal{B}} \|J_{\text{avg}}(M)\|^2,
> >     \quad \quad
> > J_{\text{avg}}(M)=\frac{1}{|M|}\sum_{(\mathbf{x}, y) \in M} (\nabla_{\boldsymbol\theta} \mathbf{z})(\frac{1}{C}\mathbf{1})$$
> >
> > Unit Jacobian: $$\mathcal{L} _ {\mathcal{B}}(\boldsymbol\theta) + \lambda \frac{|M|}{|\mathcal{B}|}\sum_{M \subseteq \mathcal{B}} \|J_{\text{unit}}(M)\|^2,
> >     \quad \quad
> > J_{\text{avg}}(M)=\frac{1}{|M|}\sum_{(\mathbf{x}, y) \in M} (\nabla_{\boldsymbol\theta} \mathbf{z})(\mathbf{u}) ,$$
> >
> > Where $\mathbf{u}$ is a random vector from the unit hypersphere and $\nabla_{\boldsymbol\theta} \mathbf{z}$ is the Jacobian of the network. In words, the Average Jacobian case can be thought of as penalizing the norm of the Jacobian with equal and uniform weighting on each class (penalizing the Jacobian directly is not computationally practical as it is itself already a matrix). In the Unit Jacobian case, we essentially are penalizing the Jacobian where the weighting on each class is random but sums to one. Additionally, recall from Sections 2.1 and 2.2 that the Gradient Norm and Fisher Trace regularizations can be viewed as penalized the product of the Jacobian and a *loss-output*  gradient $(\nabla_{\mathbf{z}}\ell(\mathbf{x}, y; \mathbf{\theta}))$ (i.e. the gradient of the loss with respect to the *output* of the model):
> >
> > Gradient Norm: $$\mathcal{L} _ {\mathcal{B}}(\boldsymbol\theta) + \lambda \frac{|M|}{|\mathcal{B}|}\sum_{M \subseteq \mathcal{B}} \|\nabla_{\boldsymbol\theta} \mathcal{L} _ {M}(\boldsymbol\theta)\|^2, \quad \quad \nabla_{\boldsymbol\theta} \mathcal{L} _ {M}(\boldsymbol\theta) = \frac{1}{|M|}\sum_{(\mathbf{x}, y) \in M} (\nabla_{\boldsymbol\theta}\mathbf{z}) (\nabla_{\mathbf{z}}\ell(\mathbf{x}, y; \mathbf{\theta}))$$
> >
> > Fisher Trace:  $$\mathcal{L} _ {\mathcal{B}}(\boldsymbol\theta) + \lambda \frac{|M|}{|\mathcal{B}|}\sum_{M \subseteq \mathcal{B}} \|\nabla_{\boldsymbol\theta} \widehat{\mathcal{L}} _ {M}(\boldsymbol\theta)\|^2, \quad \quad \nabla_{\boldsymbol\theta} \widehat{\mathcal{L}} _ {M}(\boldsymbol\theta) = \frac{1}{|M|}\sum_{\mathbf{x} \in M, \hat{y} \sim p_{\boldsymbol\theta}(y \mid \mathbf{x})} (\nabla_{\boldsymbol\theta}\mathbf{z}) (\nabla_{\mathbf{z}}\ell(\mathbf{x}, \hat{y}; \mathbf{\theta}))$$
> >
> > Thus, the only real difference between each regularization method is the choice of vector that the Jacobian of the network is taken the product with. Notably in the latter two cases, the loss-output gradient is input-dependent, which means it may not be the same from image to image and internally may vary considerably on the weighting of each class. As this is the only difference between each method, and since both Jacobian regularizations do not use loss-output gradients, the author team highlights that using some form of loss-output gradient may actually be crucial for recovering SGD generalization performance, whether by regularizing itself on its own and/or stabilizing the penalization of the Jacobian.
> >
> > We have since modified this paragraph in our updated draft in order to make this idea more clear.

---

### Official Review · Reviewer_QhHW · 2022-10-26

**Confidence:** 4
**Correctness:** 3
**Technical Novelty And Significance:** 2
**Empirical Novelty And Significance:** 3
**Recommendation:** 6

**Clarity, Quality, Novelty And Reproducibility:**

Clarity:
The paper was mostly clear

Quality:
The quality of the work is reasonably high

Novelty:
The novelty of the submission is limited, but this is reasonable for an empirical study

Reproducibility:
It would be relatively straightforward to replicate the results.

Other comments:
1) I do not understand point three in the list of main contributions. Please could the authors clarify what they mean?
2) Note that all of the mechanisms in the first paragraph of section 2 are closely related.
3) Section 2.1 first paragraph: Barrett and Dherin study full batch gradient descent, the correct citation here is Smith et al., which extended Barrett and Dherin's results to SGD.

**Strength And Weaknesses:**

Strengths:

1) The micro-batch concept enables the authors to directly minimize the modified loss proposed in Smith et al. while increasing the batch size. This is also practical, since large batch training is usually achieved by parallelizing across multiple devices.
2) The authors provide experiments across a range of models/datasets, which consistently show that gradient norm and Fisher trace regularization closes the gap between small and large batch training, thus supporting the claim that small batch SGD implicitly regularizes these quantities.
3) The authors confirm empirically that these regularizers perform poorly when the micro-batch size is too large.

Weaknesses:
1) The novelty of the work is quite limited, however I think this is fine for an empirical evaluation.
2) It would have been nice to include an ImageNet experiment.
3) I didn't understand how the grafting experiments connected to the main theme of the paper?
4) A longer background section or appendix describing the relevant prior work in detail would help readers less familiar with the topic.

**Summary Of The Paper:**

The paper provides an empirical study comparing a range of proposed mechanisms by which small batch SGD enhances generalization.

To achieve this, the authors split large mini-batches into a set of "micro-batches". They then evaluate a range of proposed implicit regularization terms on each micro-batch before summing across micro-batches to form a single large batch update. They show that applying the gradient norm/Fisher trace regularizers proposed by Smith et al. [2021] and Jastrzebski et al. [2020] to these micro-batches consistently closes the generalization gap between small and large batch training, while Jacobian regularization does not. Additionally, as predicted by prior work, they confirm that the benefits of gradient norm regularization diminish as the micro-batch size rises.

**Summary Of The Review:**

The authors present some useful experiments clarifying which proposed mechanisms for the implicit regularization of small mini-batches are most effective in practice at closing the generalization gap between small and large batch training.

I am currently recommending marginal accept since I think this is a helpful contribution. The experiments are reasonably thorough but no large scale datasets are included, and the novelty/originality of the submission is not very high.

---

> ### Author Response · Authors · 2022-11-16
> **Reviewer Response**
>
> We thank the reviewer for their response! Below, we address some of the concerns brought up:
>
> >**It would have been nice to include an ImageNet experiment.**
>
> We agree with the reviewer that experiments using the full ImageNet-1k would be an interesting point of reference. However, we note that training on full ImageNet-1k dataset is impractical given our compute resources due to the costly nature of calculating the Hessian at each training iteration. Thus, we restricted our experiment to *TinyImageNet* in order to somewhat work with the ImageNet dataset. We note that in similar papers that we cite as complementary work (namely Jastrzebski et al.) also make this choice of using TinyImageNet rather than ImageNet.
>
> >**I didn't understand how the grafting experiments connected to the main theme of the paper?**
>
> We apologize for any lack of clarity with regards to the grafting experiments, and will attempt to clarify here. The reviewer is right that the grafting experiments are somewhat orthogonal to the main theme (i.e. implicit regularization in SGD); instead, the grafting experiments are included to highlight two major points:
> - When we look at *other* solutions to improving SGD generalization, we find that mimicking the SGD micro-batch gradient norm behavior is **not** a necessary condition for bridging the small batch/large batch generalization gap (as in some cases the grafting experiments recover performance but have noticeably different norm behavior).
> - The starkly different results on the grafting experiments when varying the dataset from CIFAR10 to CIFAR100 highlight the wider epistemic issue that seemingly benign dataset choices in deep learning papers may have an outsized effect on the conclusions drawn in such projects.
>
> We have attempted to make the reasoning behind these experiments more clear in our updated draft.
>
> > **I do not understand point three in the list of main contributions. Please could the authors clarify what they mean?**
>
> We apologize for the lack of clarity here. Point three is tied to section 4.1 on the performance at large micro-batch sizes. Recall that in our initial experiments that successfully close the generalization gap, the micro-batch size is a small fraction of the total batch size (i.e. 128 of 5,120). However, in this section, we show that if the micro-batch size now is closer to the actual batch size (for instance, 2560 of 5120), then the generalization benefits of both successful regularization mechanisms no longer hold. Additionally, we see that in this regime, the micro-batch norm behavior now no longer mimics the SGD generalization behavior (as it did when the micro-batch size was small). We have since edited this section of the main contributions in order to be more clear.
>
> >**Note that all of the mechanisms in the first paragraph of section 2 are closely related.**
>
> The author team would like to note that though all of the mechanisms mentioned in Section 2 paragraph 1 are similar by definition (as they all investigate the question of generalization with SGD), the mechanisms employed do vary significantly across papers. Namely, the papers cited include analyses just related to the learning rate-to-batch size ratio, the “flat minima” theory, SGD as a signal with additive noise, SGD as a signal with multiplicative noise, and the array of implicit regularization mechanisms proposed that are the focus on the paper.
>
> > **Section 2.1 first paragraph: Barrett and Dherin study full batch gradient descent, the correct citation here is Smith et al., which extended Barrett and Dherin's results to SGD.**
>
> We thank the reviewer for this note, and have such fixed the citation in the paper.

---

> > ### Comment · Reviewer_QhHW · 2022-11-23
> > **thanks for your response**
> >
> > Thank you for your response. A few quick comments:
> >
> > 1) I think it should be possible to evaluate gradient norm regularizers without explicitly instantiating the Hessian: eg in Jax one can just directly take the gradient of the norm of a gradient. This might be harder in other frameworks.
> >
> > 2) I personally still feel that the grafting experiments do not fit well in the paper, but I note that other reviewers appreciated these so am happy to let this be! I'm not sure I follow the logic of what we learn by mimicking the evolution of the gradient norms (since directly regularizing the gradient norm is not equivalent to simply matching its scale).
> >
> > 3) Thank you for clarifying point 3. Note that this breakdown of gradient/fisher norm regularization at large mini/micro-batch size is predicted in prior work (since it can be seen straightforwardly by evaluating the expectation value of the mini/micro-batch gradient norm).
> >
> > 4) To clarify my comment on section 2, paragraph 1. I agree that these mechanisms seem superficially different, but underneath the connections between them are strong. For example, gradient norm regularization regularizes the empirical fisher, which introduces a bias towards minima with low non-uniformity, and additionally the strength of this regularization is proportional to the ratio of the learning rate to the batch size.

---

### Author Response · Authors · 2022-11-18
**General Response**

We thank the reviewers for their detailed comments, suggestions and feedback.

We are happy to see the positive reception of our work and are encouraged by the fact that many of the reviewers find our work to be of a reasonably high quality (Reviewer QhHW), and find our results to be interesting (Reviewers i18E, rQEx and fWma) and of potential (Reviewers rQEx).

In addition to addressing each reviewer’s individual concerns in their respective threads, we would also like to briefly address some common concerns, as well as highlight larger changes to the paper.
- **Primary contribution of our work**: We emphasize that our primary contribution is a comparative review of multiple proposed regularization mechanisms for closing the small- vs. large-batch generalization gap. Comparing these regularization mechanisms and understanding which ones perform better (and where) can help shed light on the training dynamics of small-batch SGD.
- **Confusion between small-batch and micro-batch size**: We would like to clarify that *small-batch* refers to the small minibatch size used for SGD. We emphasize that a “micro-batch” is different from a small mini-batch (or any minibatch). More specifically, micro-batches are subsets of a given batch (i.e., of a given mini-batch). We usually refer to microbatches in the context of large mini-batch SGD, where gradients are accumulated to preserve memory.
- **Experiments with standard SOTA optimization modifiers**: Based on Reviewer i18E’s suggestion, we have added an additional brief section of experiments in Appendix A.4 testing whether the explored regularization mechanisms work in the presence of typical SGD modifications like momentum, weight decay, and learning rate scheduling. Here, we show that our results hold in the presence of these modifiers, as the relative performance of each training setup remains intact.

We have updated our manuscript to reflect the clarifications above. We once again thank the reviewers for their detailed comments, suggestions, and feedback!

---

### Decision · Program_Chairs · 2023-01-20

**Decision:**

Accept: poster

**Justification For Why Not Higher Score:**

The reviewers highlighted several shortcomings of the paper, including low overall performance of some models, lack of large-scale experiments, and an insufficient discussion of related work. These limitations are either understandable given the scope and goals, or were addressed in the discussion phase. The major weakness that persists is the lack of novelty relative to prior work, as the paper is basically performing an improved/collected empirical investigation of ideas that have appeared previously. Therefore, while this paper makes important contributions, it does not break much new ground and that is why I do not recommend a higher score.

**Justification For Why Not Lower Score:**

The reviewers and I agree that the empirical investigation is interesting with useful conclusions that are relevant for the ICLR community. It addresses salient questions and is written clearly. There is significant value in this work and for that reason I do not recommend a lower score.

**Metareview: Summary, Strengths And Weaknesses:**

This paper undertakes an empirical investigation of several mechanisms that have been proposed as explanations for the implicit regularization effects of small-batch SGD. By applying different regularization techniques to subsets ("micro-batches") of the full batch, the authors conclude that some techniques (like gradient norm/Fisher trace regularization) can close the generalization gap between large and small batch training, whereas other techniques (like Jacobian regularization) do not perform as well.

The reviewers appreciated the overall motivation and experimental setup, particularly the use of micro-batches, and found the conclusions interesting and relevant to the community. The reviewers generally found the experiments to be thorough, although larger-scale experiments would have been appreciated. Some reviewers noted that the novelty is limited relative to prior work, and that that prior work was not discussed clearly or sufficiently.

Overall, the reviewers and I agree that the work is interesting and, despite some of its limitations, will likely be of interest to the ICLR community. Therefore I recommend acceptance.

**Note From Pc:**

if the above contains the word "oral" or "spotlight" please see: "oral" presentation means -> notable-top-5% and "spotlight" means -> notable-top-25%. As stated in our emails, we are disassociating presentation type from AC recommendations